

# Mining of prognosis-related genes in cervical squamous cell carcinoma immune microenvironment

Jiong Ma[1,*], Pu Cheng[1,2,*], Xuejun Chen[1], Chunxia Zhou[1] and Wei Zheng[1]

[1] Department of Gynecology, Second Affiliated Hospital, Zhejiang University School of Medicine, Hang Zhou, China
[2] Key Laboratory of Tumor Microenvironment and Immune Therapy of Zhejiang Province, Hang Zhou, China
[*] These authors contributed equally to this work.

## ABSTRACT

**Purpose.** The aim of this study was to explore the effective immune scoring method and mine the novel and potential immune microenvironment-related diagnostic and prognostic markers for cervical squamous cell carcinoma (CSSC).

**Materials and Methods.** The Cancer Genome Atlas (TCGA) data was downloaded and multiple data analysis approaches were initially used to search for the immune-related scoring system on the basis of Estimation of STromal and Immune cells in MAlignant Tumour tissues using Expression data (ESTIMATE) algorithm. Afterwards, the representative genes in the gene modules correlated with immune-related scores based on ESTIMATE algorithm were further screened using Weighted Gene Co-expression Network Analysis (WGCNA) and network topology analysis. Gene functions were mined through enrichment analysis, followed by exploration of the correlation between these genes and immune checkpoint genes. Finally, survival analysis was applied to search for genes with significant association with overall survival and external database was employed for further validation.

**Results.** The immune-related scores based on ESTIMATE algorithm was closely associated with other categories of scores, the HPV infection status, prognosis and the mutation levels of multiple CSCC-related genes (HLA and TP53). Eighteen new representative immune microenvironment-related genes were finally screened closely associated with patient prognosis and were further validated by the independent dataset GSE44001.

**Conclusion.** Our present study suggested that the immune-related scores based on ESTIMATE algorithm can help to screen out novel immune-related diagnostic indicators, therapeutic targets and prognostic predictors in CSCC.

# INTRODUCTION

Cervical squamous cell carcinoma (CSCC) is one of the most common malignancies in female reproductive system, which severely threatens female health and life quality (*Marth et al., 2018*). CSCC is highly prevalent in developing countries, accounting for 60–90%

Corresponding author
Wei Zheng, zhengwei@zju.edu.cn

of global cases (*Chen et al., 2017*). Radical hysterectomy is currently considered as the dominant therapy for early-stage cervical cancer (*Gil-Moreno & Magrina, 2019*; *Uppal et al., 2019*). With the popularization of cervical cancer screening, the therapeutic efficacy and prognosis of early-stage patients has been greatly improved (*Altobelli et al., 2019*; *Ngo-Metzger & Adsul, 2019*). Postoperative relapse and metastasis of CSCC remain the major causes of death in clinical practice (*Alvarado-Cabrero et al., 2017*; *Nanthamongkolkul & Hanprasertpong, 2018*). Patients with advanced-stage CSCC generally undergo adjuvant radiotherapy and/or chemotherapy; however, the therapeutic effect seems unsatisfactory (*Angeles et al., 2019*; *Bosque, Cervantes-Bonilla & Palacios-Saucedo, 2018*). At present, the International Federation of Gynaecology and Obstetrics (FIGO) staging classification is the major criterion for the prognostic prediction of patients with CSCC (*Matsuo et al., 2019*). Nevertheless, CSCC patients within similar clinical stage usually show diverse prognostic outcomes. Indeed, we now understand that the natural history of CSCC tumorigenesis is a continuous progress accompanied by a series of gene mutations over time. Based on this, CSCC was considered as a heterogenous collection of diseases, which were regarded as the major cause of anti-cancer treatment resistance and cancer relapse (*Bachtiary et al., 2006*; *Kidd & Grigsby, 2008*; *Srivastava et al., 2019*). As FIGO staging lacks heterogeneity of CSCC, clinical treatment decisions are now made depending on multiple factors including gene expression and mutation status other than traditional clinicopathological features. Thus, there is an urgent need to identify high-risk subgroups for individualized monitoring and optimized postoperative therapy in routine clinical practice.

Given the increasing evidence that various immune cells and inflammatory mediators are closely associated with the development of CSCC, tumor immune microenvironment is drawing accumulating attention nowadays (*Piersma, 2011*). The leukocytes, neutrophils, lymphocytes and macrophages directly contribute to the immune response, which could be easily and conveniently detected (*Chen et al., 2019*; *Heintzelman, Lotan & Richards-Kortum, 2000*; *Lu et al., 2018*; *Rangel-Corona et al., 2011*). In addition, several immune checkpoint biomarkers and cytokines have been identified to mediate the crosstalk between cancer cells and stromal microenvironment (*De Nola et al., 2019*; *Otter et al., 2019*). In the last decade, various studies have investigated the relationship between the prognosis of patients with primary CSCC and the immunological landscape through high-throughput quantitative measurements of cellular and molecular characteristics (*Minion & Tewari, 2018*; *Punt et al., 2015*). These studies revealed the great heterogeneity of the inflammatory/immune response in CSCC, which might determine to a large extent the final outcome of patients (*Bachtiary et al., 2006*). More recently, several researchers proposed a novel classification based on the immunological status of CSCC according to the ratio of different immune cells (such as monocyte/lymphocyte ratio or Th17/Treg ratio) in the tumor microenvironment, which might play a significant role in the accurate prediction of patient prognosis (*Huang et al., 2019*; *Zhang et al., 2011*). Unfortunately, almost none of the previous studies have reached clinical practice because of lacking the exploration from large sample data.

On this account, multiple immune scoring methods have been exploited using the expression data of immune-related genes in The Cancer Genome Atlas (TCGA) database which enable us to quantify the immune microenvironment status of a specific patient

(*Lee et al., 2012*; *Shen et al., 2018*). For instance, the systemic immune-inflammation index (SII) established according to peripheral lymphocyte, neutrophil and platelet counts has been considered as a good indicator reflecting the local immune response and systemic inflammation (*Fest et al., 2019*). Moreover, SII has been confirmed to have remarkable association with the prognosis of numerous tumors, including non-small cell lung cancer (*Guo et al., 2019*), esophageal cancer (*Ishibashi et al., 2018*) and colorectal cancer (*Xie et al., 2018*). However, there have been only limited studies designed to develop an immune-related prognostic signature for CSCC up to now. *Yang et al. (2019)* established a prognostic immune signature for CSCC based on differential expression analysis and LASSO penalized Cox proportional hazards regression. However, like many other immune-related models (*Cheng et al., 2016*; *Li et al., 2020*; *Lu et al., 2020*), the variables were screened out based on single gene expression or immune cell proportion. Therefore, despite the predictive efficiency of these models, their relevance to tumor immune microenvironment is still need to be demonstrated. Moreover, among several existing immune-scoring systems, which one is most suitable for CSCC is waiting to be solved.

Earlier published work by Yoshihara group presents a new algorithm that uses the transcriptional profiles of cancer samples to infer the fraction of infiltrating stromal and immune cells, called ESTIMATE (*Yoshihara et al., 2013*). Importantly, the ESTIMATE method can be broadly applied across almost all human solid cancers. Thus, the ESTIMATE method is a powerful tool for oncologists to elucidate the complex roles of tumor microenvironment and explore potential solutions for tumor heterogeneity.

To this end, our present study was designed to explore the immune scoring method suitable for CSCC. In addition, the gene members in the scoring system were further analyzed by a series of bioinformatic means to mine the novel and potential immune microenvironment-related diagnostic and prognostic markers.

## MATERIAL AND METHODS

### Database sources and pre-processing

The overall flow diagram of present study was summarized in Fig. 1. The RNA-seq counts data, SNP data, and clinical follow-up information were downloaded from the TCGA database. The Reads PerKilobase Million (FPKM) data of RNA-Seq were transformed into Transcripts PerKilobase Million (TPM) expression profiles. In consistent with previous studies, 13 metagenes (shown in ImmuneScore.genes.ids.txt, Supplemental File) corresponded to various immunocyte types, reflecting the different immune functions.

### Computational methods of multiple immune scores and result determination

The scores of each sample in the 13 types of metagenes were calculated based on the log2-transformed expression of each gene member in the immune metagene (shown in immune.meta.score.txt, Supplemental File) (*Safonov et al., 2017*). TIMER (https://cistrome.shinyapps.io/timer/) database (immune.immu.score.txt, Supplemental File) was utilized to calculate the scores of each sample in the immunocyte infiltration (six categories in total) (*Li et al., 2016*). Moreover, the ImmuneScore (that represents the

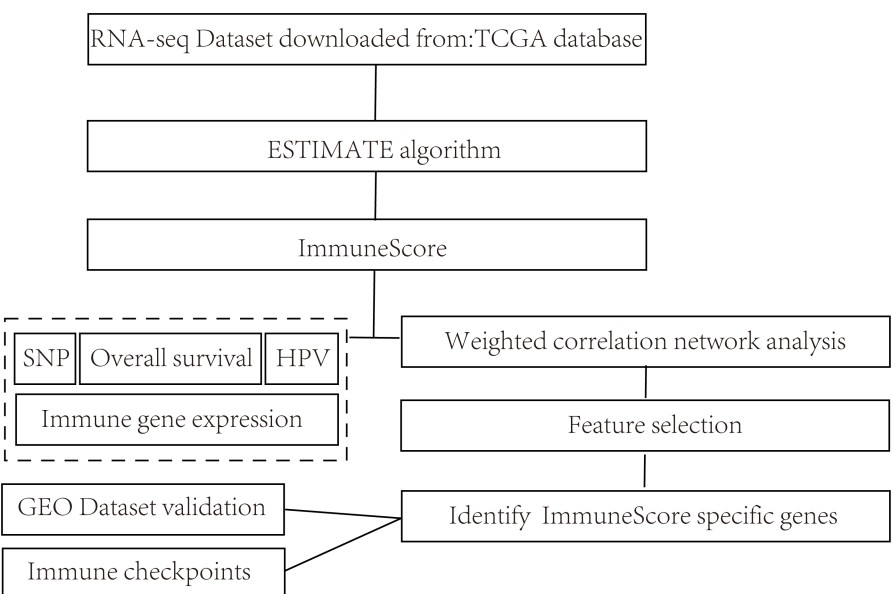

**Figure 1** Flow diagram of methods for mining of prognostic immune-related genes in CSCC.

infiltration of immune cells in tumor tissue), StromalScore (that captures the presence of stroma in tumor tissue) and ESTIMATEScore (that infers tumor purity) of each sample (immune.est.score.txt, Supplemental File) were calculated by ESTIMATE function of R software package (*Yoshihara et al., 2013*). Detailed description of above 3 scores could be found on the website below: https://bioinformatics.mdanderson.org/public-software/estimate/. Finally, R software package MCPcounter was utilized for the calculation of the abundances of ten immune-related cell (eight categories of immune cells, endothelial cells and fibroblasts) populations in the tumor microenvironment (Supplemental File).

## Survival analysis
Patients were divided into several groups according to each specific parameter (including ImmuneScore, StromalScore, ESTIMATEScore and gene expression level). Afterwards, the association between the gene expression level (or level of ImmuneScore, StromalScore, ESTIMATEScore) and overall survival was analyzed by univariate Cox regression model.

## The construction of immune scores-related gene modules through WGCNA
To begin with, transcripts with over 75% TPM of >1 and median absolute deviation (MAD) of >median were chosen from the expression profile data of all the obtained samples. Hierarchical clustering for cluster analysis of the samples was also adopted. Subsequently, samples with a distance of over 80,000 were taken as the outlier samples for screening. Moreover, the distance between any two transcripts was calculated by Pearson correlation coefficient, the establishment of the distance between any two transcripts was performed by the R software package Weighted Gene Co-expression Network Analysis (WGCNA) (*Xia et al., 2019*), and the soft threshold was set as eight for the screening of
the co-expression modules. The co-expression network has been suggested to conform to the scale-free network. In other words, the logarithm of node with the connectivity of k (log(k)) should be negatively correlated with the logarithm of the occurrence probability of the specific node (log(P(k))), and the correlation coefficient should be >0.85. Proper $\beta$ value was selected in order to ensure the network as a scale-free network. The expression matrix was subsequently transformed into the adjacent matrix, and the latter was further transformed into the topological matrix for gene clustering based on Topological Overlap Matrix (TOM) utilizing the average-linkage hierarchical clustering method in accordance with the mixed dynamic shear tree standard. In addition, the gene number of each gene network module was set at least 30. The dynamic shear method was used to determine the gene module, followed by calculation of the eigengene value of each module in succession. Afterwards, clustering analysis was performed on the modules, in which, modules close to each other were merged into a new module, with re-set appropriate height, deepSplit and minModuleSize values. Finally, the association of the acquired gene modules with ImmuneScore, StromalScore and ESTIMATEScore were separately calculated, in order to explore the gene modules with high correlation for further research.

## Establishment of the gene interaction network and functional analysis

Genes were mapped into the String database (*Szklarczyk et al., 2019*). The gene-gene interactions were acquired at the score threshold of >0.4, followed by visualization using Cytoscape software (*Shannon et al., 2003*). Meanwhile, Kyoto Encyclopedia of Genes and Genomes (KEGG) and Gene Ontology (GO) enrichment analysis was performed by utilizing the clusterprofile R package (*Xu et al., 2019*) to examine the signaling pathways affected by these genes.

## RESULTS

### The immune-related scores based on ESTIMATE algorithm is the most suitable immune scoring method for CSCC

To be specific, we retrieved CSCC samples from the TCGA database and analyzed their scores in 23 types of scoring systems, including 13 types of metagenes scores, six types of immunocyte infiltration scores, three types of immune-related scores according to ESTIMATE algorithm (ImmuneScore, StromalScore and ESTIMATEScore) and 10 types of abundances of immune-related cell. In addition, Spearman's correlation coefficient was used to calculate the correlations among these scoring systems (shown in Fig. 2). As shown in Fig. 2A, the average correlation between different types of immune-related scores was greater than 0.4. Among which, ImmuneScore ($R = 0.59$), Co_inhibition ($R = 0.59$) and LCK ($R = 0.62$) had the highest relevance with other immune scores. These findings showed that there were fine consistency and comparability between different immune scoring systems. The clustering heat maps of various types of scoring systems were shown in Fig. 2B, suggesting the great correlation among the scoring systems MHC1, MHC2, Monocytic lineage, Dendritic, Macrophages, ESTIMATEScore, ImmuneScore, Tfh, LCK, Co_stimulation, Co_inhibition, Mete_ImmuneScore, Neutrophil and STAT1. We further
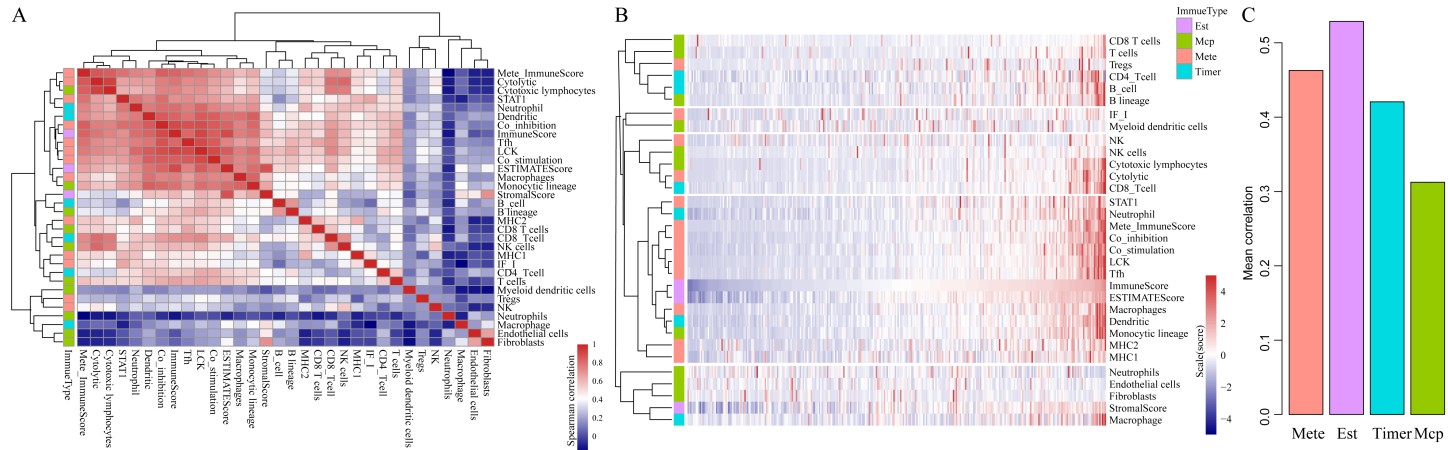

**Figure 2** **The correlations of immune-related scoring system based on ESTIMATE algorithm with other categories of scores among CSCC samples.** (A) The correlations of various immune scoring systems among CSCC samples. Spearman correlation coefficients are shown color-coded to illustrate positive (red) or negative (green) associations. (B) The clustering heat maps of various types of scoring systems. (C) The relationships among immune scores according to four different algorithms. Mete, metagene immune score; Est, ESTIMATE immune score; Timer immune score; Mcp, MCPcounter immune score.

investigated the average correlation among immune scores according to four different algorithms. As shown in Fig. 2C, the immune-related scores calculated by the ESTIMATE algorithm harbored the highest average correlation with the other three algorithms, which is greater than 0.52 on average. These findings implicated that the immune-related scores based on ESTIMATE algorithm were the most representative immune scoring methods for CSCC.

It is widely accepted that HPV infection has a significant association with the occurrence and progression of CSCC (*Ding et al., 2019*). Therefore, we separately analyzed the ImmuneScore, StromalScore and ESTIMATEScore distribution among CSCC patients with or without HPV infection. As shown in Figs. 3A–3C, the three immune-related scores in CSCC with HPV infection were significantly higher than those without HPV infection. It should be noted that ImmuneScore was most significantly correlated with the infection status of HPV ($p < 0.05$).

Subsequently, in order to investigate the association between the above three immune-related scores and prognosis, samples were sorted based on the median of scores of all samples. And then, prognostic difference was analyzed by Kaplan–Meier method (*Zou, O'Malley & Mauri, 2007*). As a result, the prognosis of samples in different groups was significantly different (shown in Fig. 4). And the five-year survival rate of samples with high ImmuneScore and ESTIMATEScore were significantly superior in comparison with those with low scores, suggesting that the three immune-related scores on the basis of ESTIMATE algorithm could be accepted as promising novel prognostic markers for CSCC.

A large number of somatic mutations of HLA genes have been reported in CSCC, strongly indicating that loss of function due to HLA mutations is tightly correlated with the immune escape of cancer cells (*Xiao et al., 2013*). It is of great significance for us to analyze
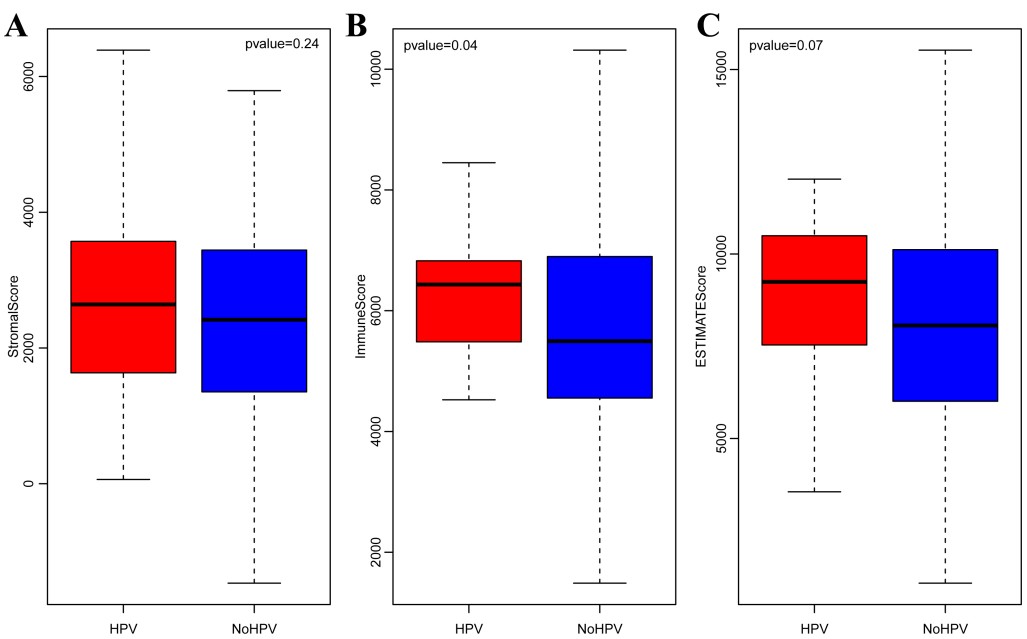

**Figure 3** **StromalScore (A), ImmuneScore (B) and ESTIMATEScore (C) distribution among CSCC patients with or without HPV infection.**

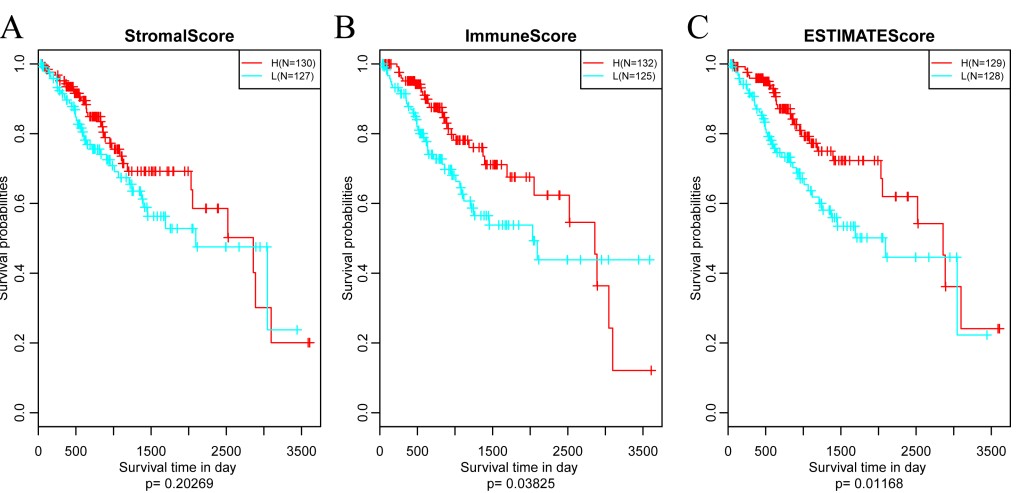

**Figure 4** **The relationships between levels of StromalScore (A), ImmuneScore (B) or ESTIMATEScore (C) and prognosis for CSCC patients.** H, High immune score; L, Low immune score.

the changes of HLA gene sequence in tumor patients. In addition, the mutation of TP53, a tumor suppressor gene, can induce unlimited proliferation and apoptosis resistance of tumor cells (*Laprano et al., 2014*; *Li et al., 2015*). Next, we focused on analyzing the associations of three immune-related scores with mutations of HLA and TP53. To this end, we extracted the mutation data of HLA-A, HLA-B, HLA-C and TP53 from the mutect-processed SNP database and then calculated the three immune-related scores based on

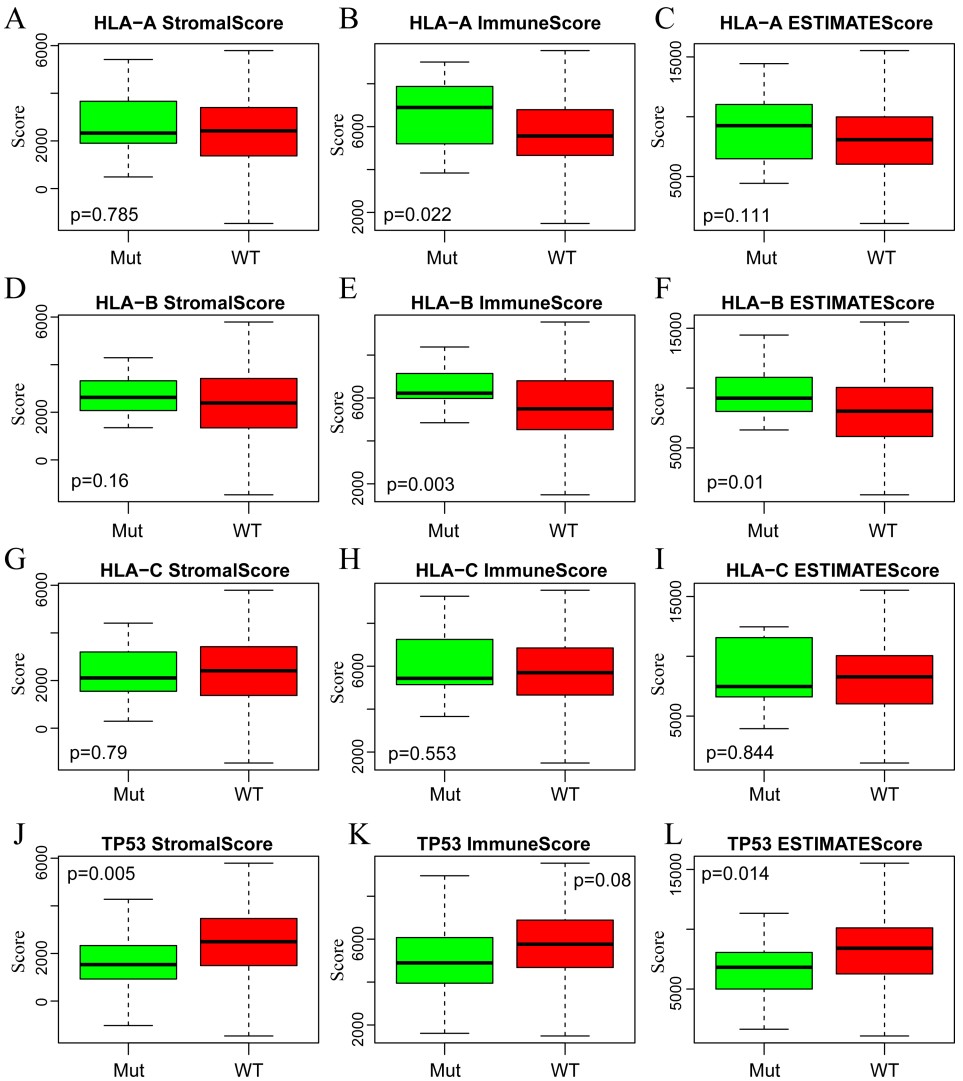

**Figure 5** **The correlations of immune-related scores based on ESTIMATE algorithm with gene mutations.** The StromalScore (A, D, G, J), ImmuneScore (B, E, H, K) and ESTIMATEScore (C, F, I, L) were calculated respectively in HLA-A (A, B, C), HLA-B (D, E, F), HLA-C (G, H, I) and TP53 (J, K, L) mutation and non-mutation groups. Green represents the mutant group and red represents the wild type. Mut, Mutant; WT, Wild type.

ESTIMATE algorithm in HLA-A, HLA-B, HLA-C and TP53 mutation and non-mutation groups. As shown in Fig. 5, there was higher level of ImmuneScore in HLA-A and HLA-B mutation groups compared with wild-type groups, while there was also higher level of ESTIMATEScore in HLA-B mutation groups but lower level in TP53 mutation groups comparison with that in wild-type groups.

In summary, we demonstrated that the immune-related scores on the basis of ESTIMATE algorithm were the most proper immune scoring method for CSCC. Additionally, the co-expressed genes with remarkable correlation with these three immune-related scores might be considered as the representative genes in CSCC immune microenvironment, which
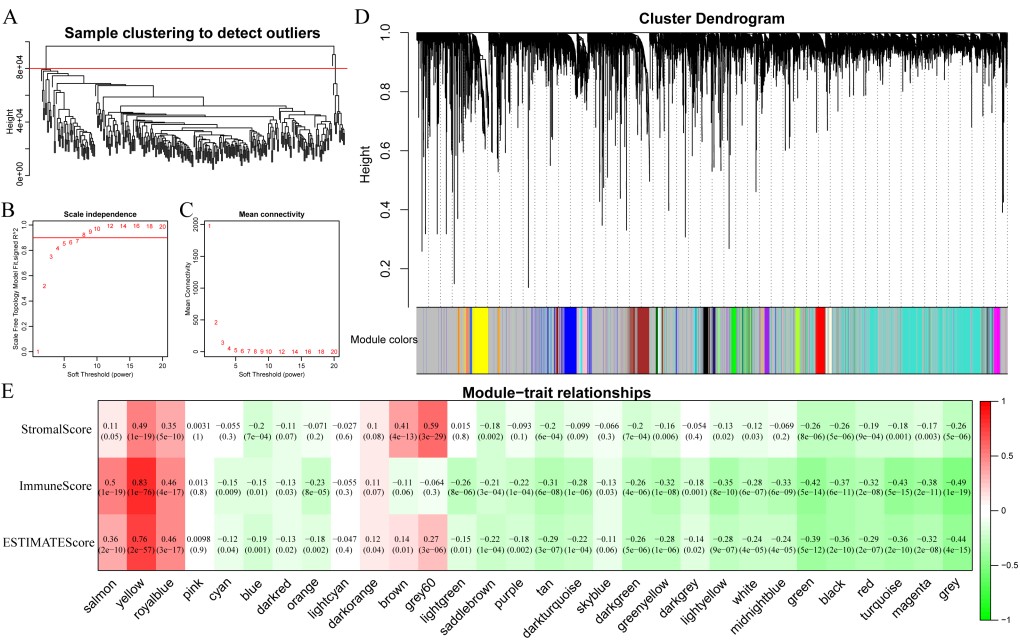

**Figure 6** **Immune scores-related gene modules mined through WGCNA.** (A) Sample clustering analysis. (B, C) Analysis of network topology for various soft-thresholding powers. (D) Gene dendrogram and module colors. (E) Correlation between each module and three immune-related scores.

could be further validated as potential prognostic markers and novel therapeutic targets of CSCC.

## Screening of the representative genes in the immune scores-related gene modules

In this section, clustering analysis was first conducted through hierarchical clustering. As shown in Fig. 6A, a total of 296 samples were finally screened out among all the outlier samples, which had a distance of larger than 80,000. Subsequently, the weight co-expression network was constructed by WGCNA with $\beta$=8 to guarantee the scale-free network (Figs. 6B and 6C). Afterwards, dynamic shear method (*Dong & Horvath, 2007*) was utilized to determine the gene modules, and clustering analysis was performed on these modules. Additionally, modules with close distance were further merged into the new module, having height, deepSplit and minModuleSize set to 0.25, 2 and 30, respectively. Finally, a total of 30 modules were acquired (Fig. 6D). Of note, the grey module indicated gene sets that could not be clustered into other modules. The transcripts of each module were counted and displayed in Table 1. In total, 6,679 transcripts were allocated to 29 co-expression modules. The correlations of the eigenvectors of these 30 modules with ImmuneScore, StromalScore and ESTIMATEScore were subsequently calculated, respectively. As shown in Fig. 6E, the yellow module obviously harbored extremely high association with these three immune-related scores based on ESTIMATE algorithm containing 422 genes.

The gene functions in the yellow module were subsequently analyzed. Meanwhile, KEGG and GO enrichment analysis was also conducted using the clusterProfiler of R software

**Table 1  Number of transcripts in each module.**

| Modules | Genes |
| --- | --- |
| Black | 232 |
| Blue | 676 |
| Brown | 469 |
| Cyan | 82 |
| Darkgreen | 53 |
| Darkgrey | 44 |
| Darkorange | 38 |
| Darkred | 54 |
| Darkturquoise | 47 |
| Green | 276 |
| Greenyellow | 110 |
| Grey | 7,417 |
| Grey 60 | 67 |
| Lightcyan | 68 |
| Lightgreen | 66 |
| Lightyellow | 65 |
| Magenta | 181 |
| Midnightblue | 78 |
| Orange | 40 |
| Pink | 232 |
| Purple | 116 |
| Red | 261 |
| Royalblue | 64 |
| Saddlebrown | 31 |
| Salmon | 97 |
| Skyblue | 33 |
| Tan | 98 |
| Turquoise | 2,642 |
| White | 37 |
| Yellow | 422 |

package, with flase discovery rate (FDR) set as <0.05. The detailed enrichment results were shown as supporting information file (yellow enrich.txt). As a result, the genes in the yellow module were enriched into 50 KEGG pathways, 670 GO biological processes (BP), 85 GO cellular components (CC) and 74 molecular functions (MF). The most significant top 20 KEGG pathways and GO terms were shown in Fig. 7. The enriched pathways mainly included Th1 and Th2 cell differentiation, cytokine-cytokine receptor interaction and so on. And the enriched biological processes primarily included T cell activation, leukocyte cell–cell adhesion and so on. The enriched cell components mainly included MHC class II protein complex and T cell receptor complex, and so on. The enriched molecular functions mainly included cytokine receptor activity and MHC class II receptor activity, and the rest. Intriguingly, these enriched pathways and GO term have previously been reported to have
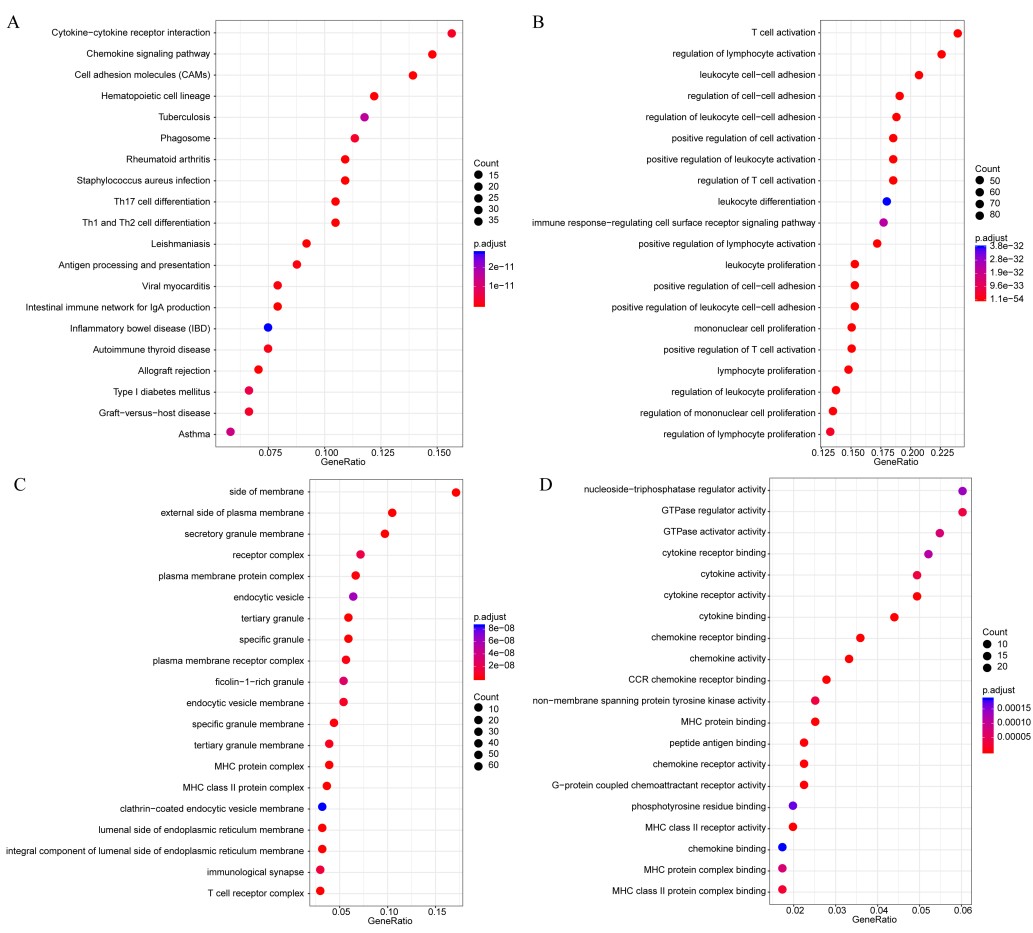

**Figure 7  The KEGG pathway and GO enrichment analysis of the genes in yellow module.** (A) Top 20 KEGG pathways enriched by the genes in yellow module. (B) Top 20 GO BP terms enriched by the genes in yellow module. (C) Top 20 GO CC terms enriched by the genes in yellow module. (D) Top 20 GO MF terms enriched by the genes in yellow module. GO, Gene Ontology; BP: biological process; CC, cellular component; MF, molecular function.

close association with CSCC and its immune microenvironment (*Roca et al., 2019*; *Wang et al., 2017*; *Yasmeen et al., 2010*; *Zehbe et al., 2005*).

Finally, to further mine the immune scores-related genes, the weight co-expression relationship between genes in the yellow modules was calculated, with the weight threshold greater than 0.2. Cytoscape software was used for derivation and visualization of the co-expression network of these genes (as shown in Fig. 8A). Afterwards, we further analyzed the topological properties of the network, which contained 244 nodes and 4,083 edges, indicating that genes with greater association with modules had more close correlation with other genes in the network. As shown in Fig. 8B, the degree distribution of the network was further analyzed, suggesting that the degree of the majority of nodes was extremely small, while the degree of a few nodes was rather large, which was consistent with the characteristics of biological network. The correlation between the gene and the module

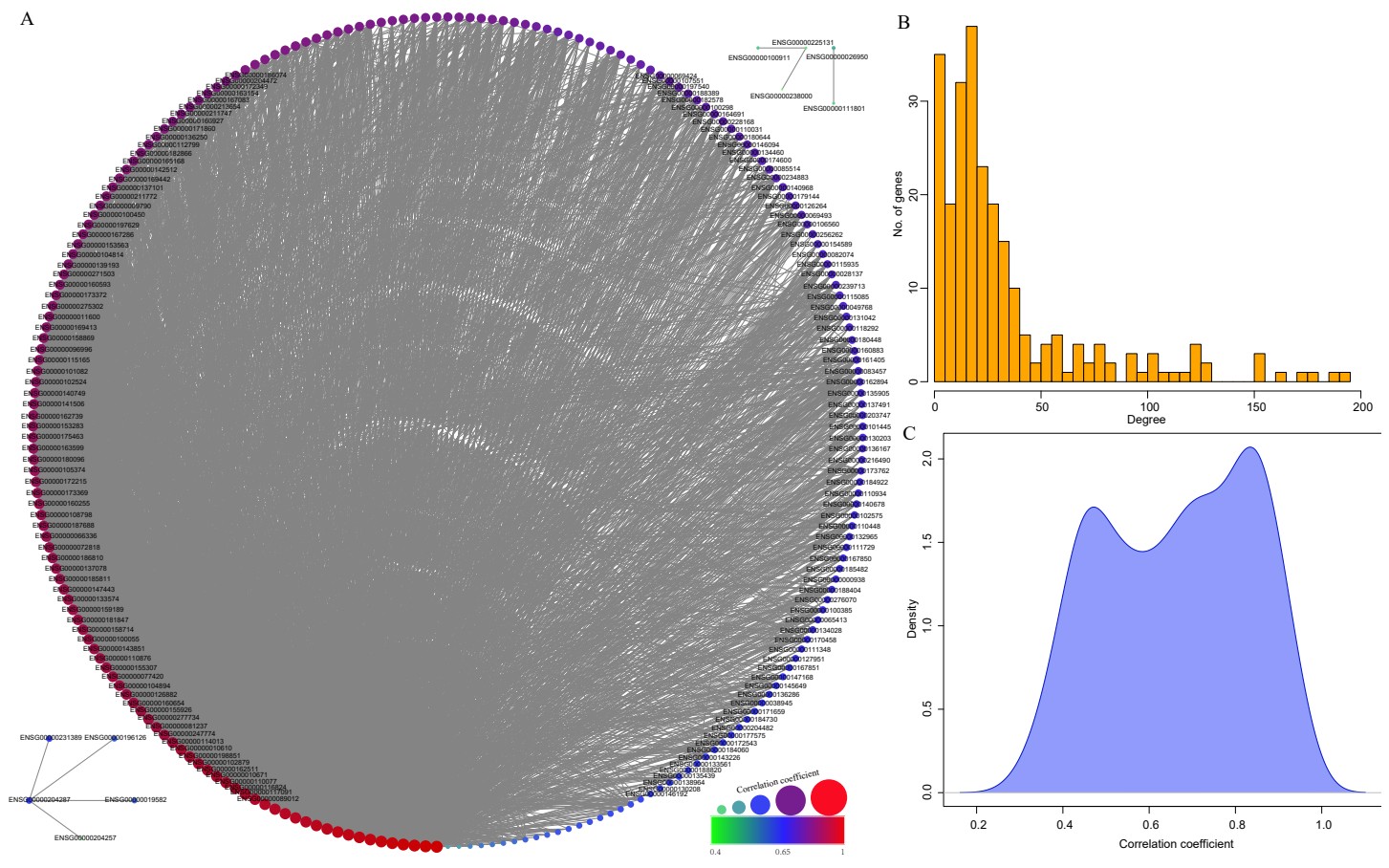

**Figure 8** **Construction of co-expression network of yellow module-related genes.** (A) Co-expression network of weights between genes in yellow module. (B) The degree distribution of nodes in yellow module. (C) The correlation of genes and module in the network.

was further calculated. As shown in Fig. 8C, the correlation between most genes and the module was over 0.6, suggesting a high expression similarity between the genes in the module. Moreover, a total of 26 genes (Table 2 and lst.genes.txt as Supplemental File) with a correlation over 0.9 and a degree over 50 in the network were selected, with seven members of LCK Metagenes, and one member of Co_inhibition Metagenes. Thus, 18 new representative immune microenvironment-related genes were finally screened.

## Function analysis of 18 novel representative immune microenvironment-related genes in CSCC patients

Firstly, to further analyze the functions of these 18 novel representative immune microenvironment-related genes, the R software package clusterProfiler was utilized for KEGG and GO enrichment analysis, with the significance FDR set at <0.05. The detailed results were summarized in lst enrich.txt (Supplemental File). In brief, these 18 genes were enriched into 11 KEGG pathways, 202 GO biological processes, 8 GO cell components, 19 molecular functions. The most significant 20 KEGG pathways and GO terms were shown in Fig. 9, the majority of which were involved in the proliferation, growth and

**Table 2   Genes with a correlation over 0.9 and a degree over 50 in the network.**

| ENSG | Symbol | corr.R | Degree | MeteGene |
|---|---|---|---|---|
| ENSG00000015285 | WAS | 0.964019 | 188 | |
| ENSG00000110324 | IL10RA | 0.944217 | 154 | LCK |
| ENSG00000134516 | DOCK2 | 0.932541 | 113 | |
| ENSG00000149781 | FERMT3 | 0.957826 | 171 | |
| ENSG00000043462 | LCP2 | 0.941048 | 102 | CLK |
| ENSG00000185862 | EVI2B | 0.94047 | 153 | LCK |
| ENSG00000117091 | CD48 | 0.918649 | 107 | LCK |
| ENSG00000089012 | SIRPG | 0.918974 | 119 | |
| ENSG00000135077 | HAVCR2 | 0.932432 | 95 | Co_inhibition |
| ENSG00000116824 | CD2 | 0.915954 | 124 | LCK |
| ENSG00000142347 | MYO1F | 0.962694 | 193 | |
| ENSG00000198851 | CD3E | 0.90917 | 130 | |
| ENSG00000123329 | ARHGAP9 | 0.925285 | 126 | |
| ENSG00000010671 | BTK | 0.913087 | 85 | |
| ENSG00000105122 | RASAL3 | 0.92036 | 124 | |
| ENSG00000162511 | LAPTM5 | 0.912211 | 72 | |
| ENSG00000005844 | ITGL | 0.92691 | 125 | |
| ENSG00000010610 | CD48 | 0.908066 | 57 | |
| ENSG00000123338 | NCKAP1L | 0.953232 | 162 | |
| ENSG00000102879 | CORO1A | 0.909449 | 94 | LCK |
| ENSG00000126860 | EVI2A | 0.923238 | 70 | |
| ENSG00000143119 | CD53 | 0.957049 | 178 | LCK |
| ENSG00000160791 | CCR5 | 0.926682 | 104 | |
| ENSG00000110077 | MS4A6A | 0.915621 | 57 | |
| ENSG00000122122 | SASH3 | 0.949558 | 153 | |
| ENSG00000167208 | SNX20 | 0.942276 | 123 | |

differentiation of T cells. Intriguingly, LAPTM5, EVI2A and MS4A6A were not enriched in any signaling pathways and GO term, indicating that the functions of these three genes remained completely unclear, which is the focus of our further studies.

Secondly, to further investigate the potential roles of the 18 novel representative immune microenvironment-related genes in clinical practice, the R package corrgram was utilized for the calculation of the association between these genes and immune checkpoints (PDCD1, CD274, PDCD1LG2, CTLA4, CD86, CD80, CD276, VTCN1). As shown in Fig. 10, apart from CD276 and VTCN1, the other 6 immune checkpoints were significantly related to these 18 genes, with an average correlation coefficient over 0.5, which indicated that these immune microenvironment-related genes might be promising targets for immunotherapy.

Finally, the prognostic significance of 18 novel representative immune microenvironment-related genes was assessed. According to the median of gene expression, samples were categorized into high and low expression groups. And then the differences of prognosis between these groups were analyzed. As shown in Fig. 11, high expression of 13 genes were

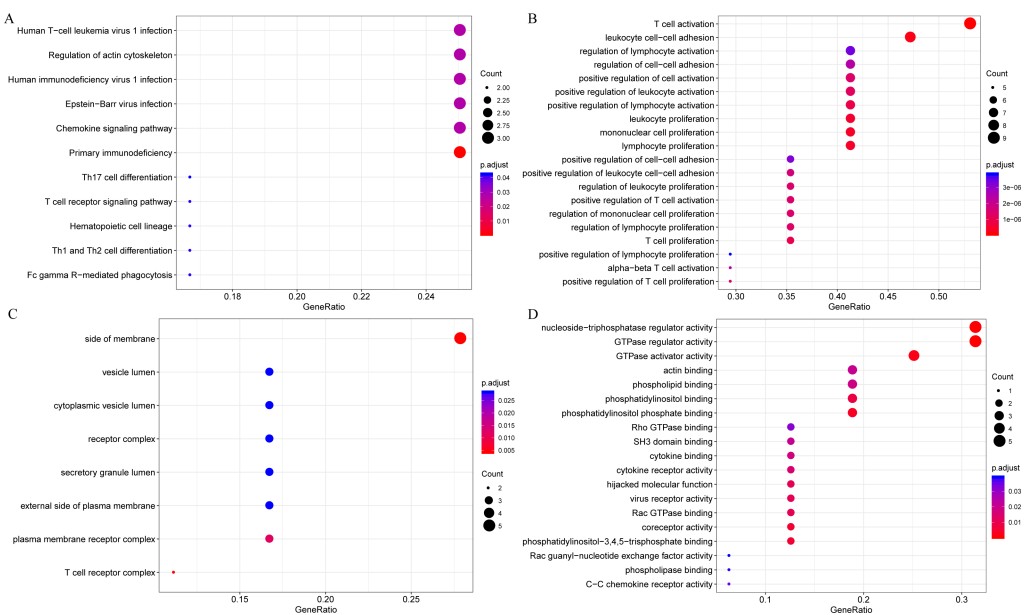

**Figure 9** **The KEGG pathway and GO enrichment analysis of 18 novel representative immune microenvironment-related genes for CSCC patients.** (A) Top 20 KEGG pathways enriched by 18 novel representative immune microenvironment-related genes. (B) Top 20 GO BP terms enriched by 18 novel representative immune microenvironment-related genes. (C) Top 20 GO CC terms enriched by 18 novel representative immune microenvironment-related genes. (D) Top 20 GO MF terms enriched by 18 novel representative immune microenvironment-related genes. GO, Gene Ontology; BP, biological process; CC, cellular component; MF, molecular function.

significantly associated with better overall survival according to the threshold of $p < 0.05$, suggesting that these genes might be closely associated with patient prognosis.

**Validation of the correlations of 18 immune microenvironment-related genes with ImmuneScore for CSCC patients by using external dataset**

External database was used for further validation of the correlations of 18 immune microenvironment-related genes with the immune-related scores according to ESTIMATE algorithm for CSCC patients. Standardized expression matrix was downloaded and extracted from an independent dataset GSE44001 (*Lee et al., 2013*) from Gene Expression Omnibus (GEO). R packages hgu133plus2.db was utilized to map a probe for gene to extract the expression profiles of these 18 genes, followed by the calculation of the ImmuneScore for each sample using R software package ESTIMATE. Subsequently, the Pearson correlation was calculated between expression of these genes and the level of ImmuneScore for every CSCC sample in this dataset. As shown in Fig. 12, apart from CCR5 ($P = 0.867$, $R = 0.01$), the other 17 genes were significantly associated with the ImmuneScore, which was consistent with our previous findings.

# DISCUSSION

Great attention has been paid to the association of the immune system with the pathogenesis and progression of tumor in recent years, which has shed light on CSCC therapy, promoting

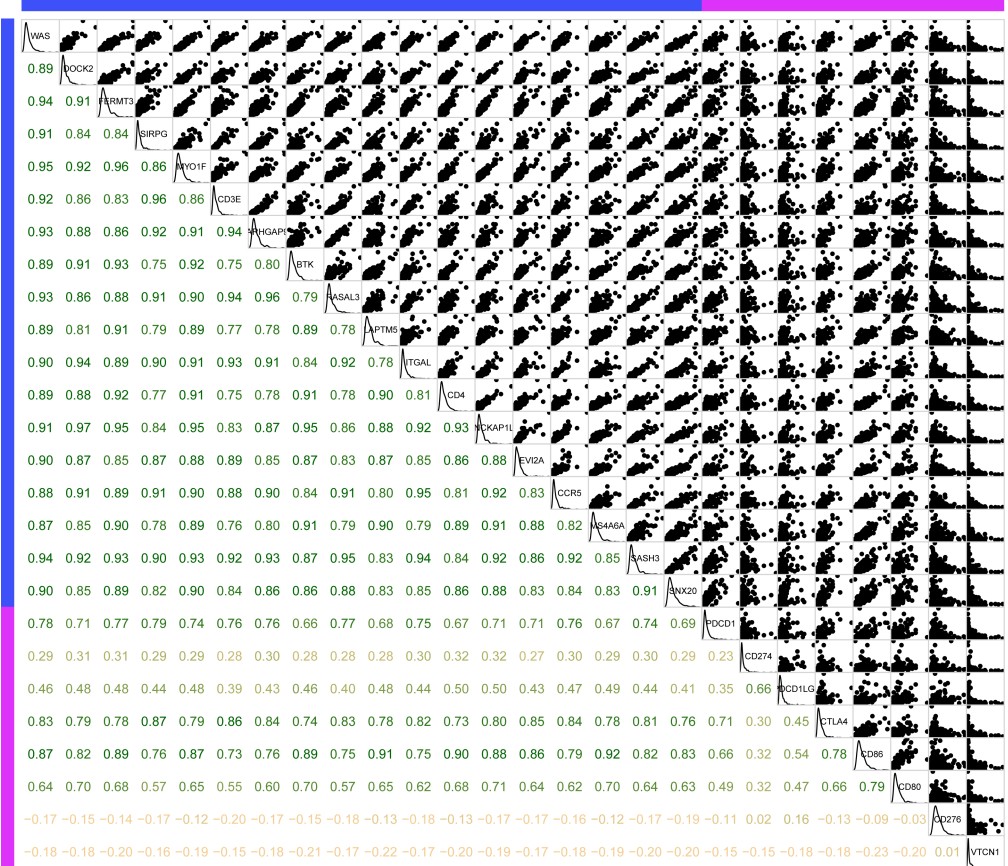

**Figure 10** **The association between 18 novel representative immune microenvironment-related genes for CSCC patients and immune checkpoints.** Apart from CD276 and VTCN1, the other 6 immune checkpoints were significantly related to 18 immune microenvironment-related genes.

the continuous development of anti-cancer therapy (*Dyer et al., 2019*; *Orbegoso, Murali & Banerjee, 2018*). The external anti-CSCC approaches are frequently applied in previous clinical practice, including surgical resection and chemotherapy. However, the effect of surgical resection is generally restricted due to the invasion into adjacent tissues by cancer cells or distant metastasis. In addition, the application of chemotherapy is limited due to its toxicity to normal tissues (*Menderes et al., 2016*). Thus, conventional therapies would exert great burden on the body while providing therapeutic benefits. To this end, it has been widely accepted as a novel direction of anti-cancer therapy by starting from the tumor origin, in other words, the immune system of human body, to control and even kill tumor cells via the modulation of the immune system and enhancement of the anti-tumor immunity in the tumor microenvironment (*Ring et al., 2017*).

The tumor microenvironment, mainly composed of immune cells, inflammatory cells, mesenchymal cells, tumor cells, stromal cells, inflammatory mediators and cytokines, provides support for tumor biological behavior including the pathogenesis, progression, invasion and metastasis (*Piersma, 2011*; *Qi & Wu, 2019*; *Tuccitto et al., 2019*). Therefore,

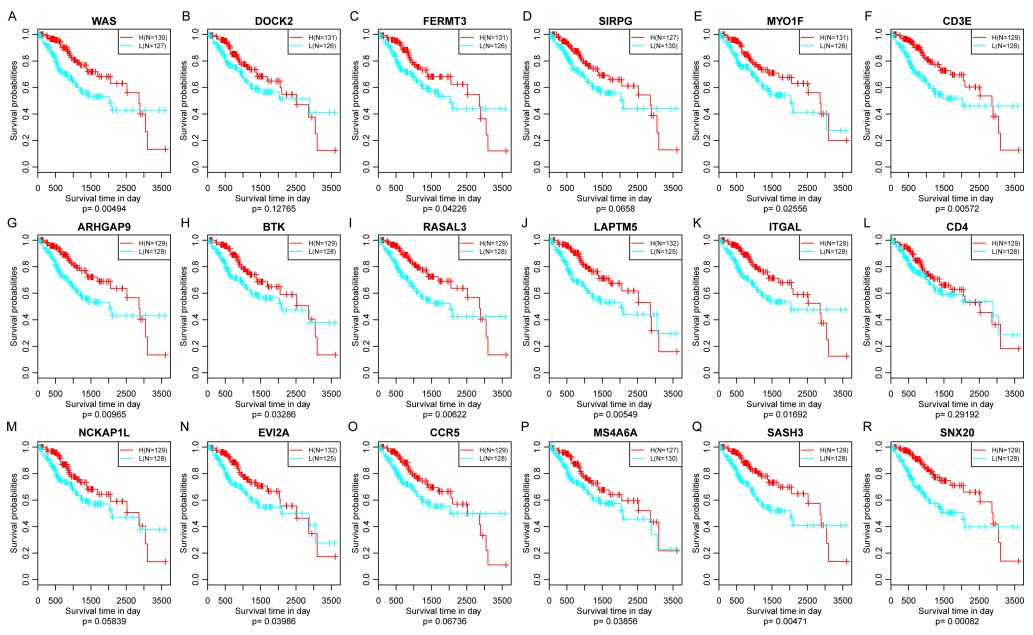

**Figure 11 The relationship between 18 novel representative immune microenvironment-related genes and prognosis (A–R).** Apart from DOCK2 ($P = 0.12765$), SIRPG ($P = 0.0658$), CD4 ($P = 0.29192$), NCKAP1L ($P = 0.12765$) and CCR5 ($P = 0.06736$), high expression of other 13 genes were significantly associated with better overall survival.

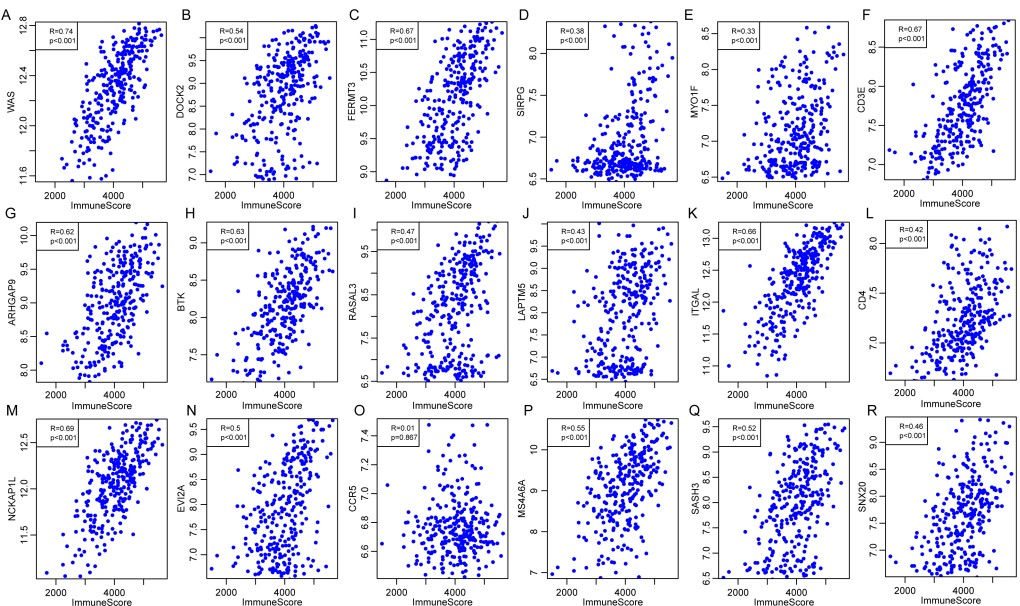

**Figure 12 The correlations of 18 immune microenvironment-related genes with ImmuneScore for CSCC patients in independent dataset (A–R).** Apart from CCR5 ($P = 0.867$, $R = 0.01$), the other 17 genes were significantly associated with the ImmuneScore.

it is of great significance to discover novel and meaningful immune microenvironment-related genes in CSCC as prognostic predictor and therapeutic targets.

In this study, the TCGA database was used to search for the immune microenvironment markers related to the survival time of CSCC patients. And 18 genes were finally detected having remarkable correlation with the prognosis of patients, which was further validated in the GEO database.

To be specific, firstly, multiple methods of data analysis were utilized to search for the three immune-related scores on the basis of ESTIMATE algorithm, showing high correlations with diverse other immune-related scores, patients prognosis, HPV infection status and the mutation levels of multiple well-defined CSCC-related genes (HLA and TP53). Secondly, the representative genes in the gene modules associated with immune-related scores according to ESTIMATE algorithm were further searched using WGCNA and network topology analysis. Thirdly, we mined the gene functions through enrichment analysis, followed by the exploration of the association between these genes and immune checkpoint genes. Finally, survival analysis was employed to search for the genes with evident correlation with OS. In addition, external database was employed for further validation of the association of these immune microenvironment-related genes with ImmuneScore for CSCC patients. In total, we successfully mined 18 novel potential immune microenvironment-related diagnostic and prognostic indicators or therapeutic targets.

Of note, 11 out of these 18 genes (IL10RA, CD4, HAVCR2, CD2, CCR5, CD3E, BTK, etc.) have previously been demonstrated to participate in the pathogenesis, progression, malignant transformation, and pathological process of immune microenvironment of CSCC, which are also significantly associated with patient survival, prognosis and diagnosis (*Cao et al., 2013*; *Che, Shao & Wang, 2016*; *Hussain et al., 2013*; *Punt et al., 2015*). These above-described observations validate the great reliability and accuracy of the bioinformatic mining results in our present study, in which, we combined TCGA database screening with GEO database for verification. However, the correlations of two genes (LAPTM5 and EVI2A) with CSCC have never been confirmed by any basic or clinical studies, which we are most interested in. LAPTM5, Laptm5, a lysosomal transmembrane protein enhancing the degradation of several targets involved in immune signaling (such as ubiquitin-editing enzyme A20), has been validated to be participate in the modulation of the lethal T cell alloreactivity mediated by dendritic cells and immunoreactions in multiple inflammatory disease, such as host versus graft disease (GVHD) (*Glowacka et al., 2012*; *Hubbard-Lucey et al., 2014*). On the other hand, EVI2A has been confirmed to be involved in lymphocyte proliferation and viability, which is a well-defined immune-specific tumor suppressor in head and neck cancer (*Li et al., 2014*).

At present, accumulating studies focus on the mining of the association of numerous genes expression with the survival of CSCC patients, however, the majority of previous studies are only performed in animal model, in vitro cell model or small sample samples of tumor patients. Thus, more comprehensive, large-scale population studies are required due to the complexity of CSCC microenvironment. Fortunately, the rapid development of genome-wide sequencing renders the free utilization of high-throughput tumor databases,

such as TCGA, making it possible to apply the bioinformatic big data for the large-scale CSCC population.

## CONCLUSION

In the present study, we mainly studied the CSCC immune microenvironment-related gene characteristics. Consequently, these genes are involved in the pathogenesis, progression and malignant transformation of CSCC, affecting OS of CSCC patients. Our present findings can offer more information to decode the complex tumor-tumor interactions in CSCC microenvironment. These findings will help to mine the novel immune-related diagnostic indicators, therapeutic targets and prognostic predictors in CSCC. Besides, the methods of our study have general applicability and provide some references value for the identification of potential diagnostic and prognostic biomarkers for other biologically heterogeneous cancers.

### Funding

This work was supported by the Natural Science Fonudation of Zhejiang Province (LY15H040007) and the Chinese National Natural Science Foundation (81902629). The funders had no role in study design, data collection and analysis, decision to publish, or preparation of the manuscript.

### Grant Disclosures

The following grant information was disclosed by the authors:
Natural Science Fonudation of Zhejiang Province: LY15H040007.
Chinese National Natural Science Foundation: 81902629.

### Competing Interests

The authors declare there are no competing interests.

### Author Contributions

- Jiong Ma conceived and designed the experiments, performed the experiments, authored or reviewed drafts of the paper, and approved the final draft.
- Pu Cheng performed the experiments, analyzed the data, authored or reviewed drafts of the paper, and approved the final draft.
- Xuejun Chen and Chunxia Zhou analyzed the data, prepared figures and/or tables, and approved the final draft.
- Wei Zheng conceived and designed the experiments, authored or reviewed drafts of the paper, and approved the final draft.

### Data Availability

The raw measurements are available as Supplemental Files.

## Supplemental Information

Supplemental information for this article can be found online at http://dx.doi.org/10.7717/peerj.9627#supplemental-information.

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
