# Peer review of "Mining of prognosis-related genes in cervical squamous cell carcinoma immune microenvironment"

_PeerJ, doi:10.7717/peerj.9627_

## Round 0.1 · original submission · Minor Revisions

Please address the critiques of both reviewers and revise your manuscript accordingly.

Reviewer 1 ·

Basic reporting

A little more background on the significance of microenvironment related diagnostic and prognostic markers would be better to provide.

Experimental design

Methods should describe in more detail why specific scores and databases mentioned are used/selected.
The reason to choose the mentioned data analysis methods to search for the three immune-related scores should be described.

Validity of the findings

How different/similar was external data set for validation of the correlations of 18 immune microenvironment related genes?
Authors should talk about the significance of their findings in more detail.

Additional comments

Figure 8: All acrony’s full form should be mentioned in the description of the figure

Reviewer 2 ·

Basic reporting

PeerJ: #45770v1
Title: Mining of prognosis-related genes in cervical squamous cell carcinoma immune microenvironment

This work reports on the applicability of the immune-related scoring approach to screen immune-related prognostic markers in cervical squamous cell carcinoma immune microenvironment. Two levels of screening were applied step by step using ESTIMATE algorithm and WGCNA and network analysis to identify prognostic genes. Considering the biological heterogeneity of cervical squamous cell carcinoma it is worth carrying out bioinformatic studies similar to this present study to contribute to the current efforts to identifying prognostic biomarkers. This work is within the scope of the journal. The authors used the English language professionally throughout the manuscript and the manuscript was appropriately structured following PeerJ standard. Figures are of publication quality and appropriately annotated. Raw data is available as supporting information files. I have pointed out minor points.

1. Cervical squamous cell carcinoma should be abbreviated as CSCC throughout the manuscript

2. Although the context of this study well-reported with sufficient citations, to aid the reader additionally, the biological heterogeneity associated with cervical squamous cell carcinoma should be discussed in details (with appropriate citations) in the Introduction section, as because the International Federation of Gynecology and Obstetrics (FIGO) staging lacks heterogeneity of CSCC

3. As this study focuses on the applicability of the immune-related scoring approach, the authors should introduce a comparative discussion (with appropriate citations) of different type of scoring approaches in the Introduction section

4. Supporting information materials should be appropriately cited throughout the manuscript (For example, Line 80, “shown in ImmuneScore.genes.ids.txt” should be written as “shown in ImmuneScore.genes.ids.txt, Supplemental File”)

5. In-text citations and references are not in PeerJ style

6. Figure 3 caption, “he relationships between levels” should be “The relationships between levels”

7. In general, all the abbreviation used should be annotated in its first appearance

Experimental design

The objective of the study is well defined and computational experiments are well structured and seem to be performed accurately following standard strategies. Authors should address the following minor issues.

1. A flowchart representing the workflow of the computational experiments performed in this work should be added in the Materials and Methods section to make the method section more comprehensive

2. A brief description of each of the ImmuneScore, StromalScore and ESTIMATEScore which is associated with the ESTIMATE algorithm should be added at appropriate place in the manuscript (maybe under “Computational methods of multiple immune scores and result determination” in the method section)

3. Line 123, appropriate citation(s) should be added for “Cytoscape software”

Validity of the findings

The results are presented stepwise in a logical manner and also in details. Statistical analyses have been performed and reported. The results are well discussed and concluded appropriately answering the original research question. There are only minor points to be taken care of.

1. Line 136, The sentence “The first three scoring systems with most obvious correlation with others including ImmuneScore (R=0.59), Co_inhibition(R=0.59) and LCK(R=0.62), indicating that the consistency among the immune scores calculated by different algorithms to a certain extent.” should be rephrased

2. Line 157, appropriate citation(s) should be added for Kaplan-Meier method

3. Line 182-183, “As shown Fig.5A,” should be “As shown in Fig.5A,”

4. Line 186, appropriate citation(s) should be added for “dynamic shear method”

5. Line 198, The sentence “The detailed enrichment results were shown in yellow enrich.txt” should be written as “The detailed enrichment results were shown as supporting information file (yellow_enrich.txt)”

6. Line 220, The sentence “Moreover, a total of 26 genes (Table 2 and lst.genes.txt)” should be written as “Moreover, a total of 26 genes (Table 2 and lst.genes.txt as supplemental file)”

7. Line 229, “summarized in lst enrich.txt” should be “summarized in lst.enrich.txt (supplemental file)”

8. Line 252, “contrib1-GPL14951.txt” file should be added as supplemental file

9. Line 258, SIRPG gene should be reported for weak association compared to other genes

10. Any future application of this work flow and applicability of this approach in mining prognosis related genes in other biologically heterogeneous carcinoma should be discussed in the discussion section

---

## Round 0.2 · accepted · Accept

Thank you for addressing all the critiques of both reviewers and for the adequate revision of the manuscript. I am please to accept your manuscript now.